# Analyses of Fatty Acids, Proteins, Ascorbic Acid, Bioactive Phenolic Compounds and Antioxidant Activity of Canadian Barley Cultivars and Elite Germplasm

**DOI:** 10.3390/molecules27227852

**Published:** 2022-11-14

**Authors:** Champa Wijekoon, Thomas Netticadan, Ali Sabra, Liping Yu, Chamali Kodikara, Ana Badea

**Affiliations:** 1Agriculture and Agri-Food Canada, Morden Research and Development Centre, Morden, MB R6M 1Y5, Canada; 2Canadian Centre for Agri-Food Research in Health and Medicine, Winnipeg, MB R3C 1B2, Canada; 3Department of Plant Science, University of Manitoba, Winnipeg, MB R3T 2N2, Canada; 4Department of Physiology and Pathophysiology, University of Manitoba, Winnipeg, MB R3T 2N2, Canada; 5Agriculture and Agri-Food Canada, Brandon Research and Development Centre, Brandon, MB R7A 5Y3, Canada

**Keywords:** elite germplasm, genotype, protein, fatty acid, ascorbic acid, phenolic compounds, antioxidant activity

## Abstract

Barley (*Hordeum vulgare* L.) grain is rich in fiber and antioxidant phytochemicals, including fatty acids, proteins, phenolic compounds, vitamins, and minerals, that offer various health benefits. Research on identifying different barley genotypes based on their health attributes is very limited. In this study, we performed an analysis of fatty acids, proteins, ascorbic acid, phenolic compounds, and antioxidant activity of several Canadian barley cultivars and elite breeding lines. Linoleic acid (C18:2) was the predominant fatty acid present in the tested barley cultivars. The cultivar CDC Bold contained the highest amount of ascorbic acid, while the highest protein content was in CDC Mindon. An assessment of the free and bound phenolic compounds of barley grains showed quantitative changes among different genotypes of Canadian barley. Catechin is the most abundant molecule in free phenolics, while ferulic acid and para-coumeric acid are the most abundant in bound phenolics. Ferulic acid and vanillic acid were molecules detected in the soluble free fraction of all genotypes. Para-coumeric acid was detected only in genotypes such as CDC Copeland, CDC Bold, Lowe, and elite breeding Line 5 of both free and bound fractions of barley. Breeding Line 5 had the lowest antioxidant activity. An analysis of the above molecules and parameters of Canadian barley would help to uncover potential biomarkers in order to distinguish individual barley genotypes.

## 1. Introduction

Barley (*Hordeum vulgare* L.) ranks fourth in global cereal production after maize, wheat, and rice [1]. The global production volume of barley was about 147.05 million metric tons in the 2021–2022 crop year, which was a decrease from around 160.53 million metric tons in 2020–2021 [2]. Barley is rich in several health-boosting components, such as dietary fiber, essential fatty acids, vitamins, and phytochemicals, including several phenolic compounds [3]. Over 60% of the total phenolic content found in beer comes from barley [4]. These barley phenolic compounds play an important role in beer quality, shelf-life, taste, and flavor [5].

Barley is a versatile crop grown for malting, feed, forage (general purpose), and food across the Canadian prairie provinces. Canada is the fourth largest producer of barley and the second largest exporter of malt barley [6]. Western Canada is a major contributor to the production and use of Canadian barley [7]. In 2021, a total of 3.257 million hectares were planted with barley and produced an estimated 6,602,000 tons, with 42.3% for feed and forage and 2.3% for human food [8].

A new stream of barley cultivars is continuously required to address new disease threats that continue to evolve at a rapid rate. In addition, new cultivars are necessary to improve the production and quality of barley, especially since it is required by the industry. The nutrient composition of the newly released cultivars and/or elite breeding lines may be different from each other or from the older cultivars. It has been reported that individual cultivars of various crops, including barley, could be distinguished based on their content of phenolic compounds [9]. Barley has higher antioxidant activity as compared with the more widely consumed cereals such as wheat and rice [10]. The quality of the barley is also determined in part by the protein content produced during grain filling, and thus the accumulation of storage proteins plays an important role in seed development [11]. The grain protein content of barley showed differences in various varieties ranging from 8 to 15% [12]. Highly significant differences were also detected in lipid content between Canadian barley cultivars, with the highest value of 3.1% [13]. Lipids, along with fatty acids, are important constituents of the barley grain and can influence the storage and processing of the grain.

Different metabolomics platforms, such as mass spectrometry coupled (MS) with liquid chromatography or gas chromatography (GC), and various bioinformatics tools have greatly facilitated the identification, isolation, structural characterization, and interpretation of biologically active compounds in nutrition research [14]. Considering the high biological activity of barley phenolic compounds, proteins, and lipids, it is important to investigate the biomolecule composition of 14 barley genotypes. The purpose of this is to uncover potential biomarkers that would aid in distinguishing individual genotypes/cultivars. Very limited research is documented on this aspect. In this study, we examined the levels of fatty acids, proteins, ascorbic acid, phenolic compounds, and antioxidant activity of several Canadian barley cultivars and elite breeding lines (Table 1).

## 2. Results

### 2.1. Comparison of Fatty Acid Composition in Selected Genotypes of Canadian Barley Grains

A percentage of fatty acid methyl esters (FAMEs) showed variation among the 14 barley genotypes (Figure 1 and Appendix A). About 15 fatty acids were detected as FAMEs and quantified as a percentage using the GC/MS analysis. For example, methyl linoleate/linoleic acid (C18:2) was the predominant constituent (47–54%) in barley, while Harrington, CDC Bow, and elite breeding Lines 2, 3, and 4 showed a significant reduction of the C18:2 content compared with the other genotypes (Figure 1 and Appendix A). The content of C18:2 in barley seeds was more than five times higher than methyl oleate/oleic acid (C18:1) content and about 10 times higher than methyl linolenate/linolenic acid (C18:3) content. Methyl palmitate/palmitic acid (C16:0) represented 24–28% composition in fatty acids and varied among the genotypes. Two compounds, citric acid trimethyl ester and methyl 9 (z), 11 (E), 13 (E) octadecatrienoate or conjugated linolenic acid (CLnA), were detected in trace amounts (Appendix A).

### 2.2. Comparative Analysis of Protein Content in Barley Genotypes

Protein content in barley showed genotypic differences. Overall, the malting barley cultivar Lowe showed the lowest content of proteins. The highest protein content was observed in the general purpose cultivar CDC Mindon (Figure 2).

### 2.3. Assessment of Ascorbic Acid Content

The ascorbic acid content was detected in the soluble extract. Genotypes of CDC Bold, CDC Copeland, CDC Mindon, Harrington, Lowe, and elite germplasm Line 4 showed significantly higher constitutive ascorbic acid content compared with most of the other genotypes analyzed (Table 2).

### 2.4. Assessment of Phenolic Compounds in Grains of Selected Canadian Barley Genotypes

Assessment of free and bound phenolic compounds of barley grains showed quantitative changes among different genotypes of barley included in the testing in this study (Table 2 and Table 3). Cultivars such as CDC Bold and CDC Bow have a relatively higher amount of free phenolic compounds than the other cultivars. Para-coumeric acid (PCA) was quantified in only four of the cultivars, CDC Bold, CDC Copeland, Lowe, and elite breeding Line 1, as it was below the limit of quantification (LOQ) in other genotypes. The cultivar Lowe showed the highest content of PCA. Free sinapic acid (SIN) contents fluctuated among the genotypes, and it was not detected in elite breeding Line 5. The cultivar CDC Bold showed relatively the highest catechin (CAT) content compared with other genotypes. Overall, free PCA, SIN, and CAT compounds were not detected in elite breeding Line 5 (Table 3 and Figure 3A).

The analysis of the bound part of phenolic compounds showed that PCA content was low in genotypes such as elite breeding Line 5 (Figure 3B), elite breeding Line 2, and CDC Bold. Caffeic acid (CAF) content was higher in CDC Mindon than in the other genotypes, such as CDC Bow, CDC Bold, and elite breeding Lines 3, 4 and 5. Sinapic acid content was significantly higher in elite breeding Line 1 and CDC Mindon compared with others such as AC Metcalfe, Lowe, and CDC Bold. Ferulic acid content was significantly higher in the Harrington genotype compared with CDC Bow, CDC Bold, and elite breeding Lines 2, 3, 4, and 5. No significant differences in the isoferulic acid (Iso FA) content were found among most of the genotypes analyzed. The elite breeding Line 5 showed significantly lower CAT content compared to those of Lowe and CDC Copeland cultivars. The content of 4-hydroxy benzoic acid (4HBA) was higher in cultivars such as AAC Synergy, AAC Goldman, and AC Metcalfe. AC Metcalfe contained high vanillic acid content, while elite breeding Line 2 and CDC Bold had the lowest. The highest VANILLIN content was in Harrington, while the lowest content was in CDC Bow and CDC Bold. Compounds such as caffeic acid, iso-ferulic acid, 4 hydroxy benzoic acid (4HBA), and vanillic acid were detected only in bound barley fractions (Table 4).

A principal component analysis of free phenolics showed that the first three principal components provided a reasonable summary of the data and explained more than 75% of the total variations of the free form of phenolic compounds in barley genotypes (Figure 4).

The analysis of free phenolics showed the first principal component had large positive associations with the FA and VANILLIN and moderate positive associations with SIN and PCA, which are responsible for the PC1. The second principal component had large positive associations with SIN and CAT, while PCA gave large negative associations. The loading plot shows the results of the first two principal components where FA and VANILLIN are the responsible phenolic acids to have a large positive loading on component 1. Therefore, the genotypes Harrington, Lowe, Line 1, and Line 3 have a positive relationship due to the presence of high FA and VANILLIN content. CAT and SIN have positive loadings on component 2, whereas genotypes CDC Bold and Line 4 have a relationship due to the presence of CAT and SIN (Figure 4A–C).

Cluster analysis generated a dendrogram, as illustrated in Figure 4D. The analysis is based on the mean values of the five measured phenolic compound concentrations in free form within 14 barley genotypes into clusters based on a similarity level of 70%. Each cluster was markedly different from the other clusters and consisted of various types of barley genotypes. The dendrogram shows there is a similarity between varieties AAC Synergy and AAC Goldman at a level of 87.41% similarity in cluster 1, and the cluster centroids show that it is because of the presence of CAT, FA, and VANILLIN. Cluster 4 has another two genotypes, CDC Copeland and CDC Mindon, which have a similarity level of 79.72% due to the positive relationship between CAT and FA. There is a 76.99% similarity level between AC Metcalfe and CDC Bow in cluster 3 based on CAT and FA. Even though cluster 3 is grouped based on the CAT and FA, the similarity is stronger in cluster 4 compared with cluster 2 (Figure 4D). Based on these analyses, AAC Synergy and AAC Goldman genotypes are highly similar, followed by genotypes CDC Copeland, CDC Mindon, AC Metcalfe, and CDC Bow.

A principal component analysis of bound phenolic compounds showed that the first three principal components provided a reasonable summary of the data and explained more than 88% of the total variations (Figure 5). The fact that variance in the dataset was distributed in the first three variables was likely due to the high correlation between the bound phenolic compounds in barley varieties which are responsible for the first three principal components. The first principal component (PC1) had large positive associations with all the analyzed bound form phenolic compounds, such as FA, PCA, CAF, SIN, Iso FA, CAT, 4HBA, and VAN A. The second principal component had large positive associations with CAT, PCA, and FA, while SIN and 4HBA gave large negative associations. The loading plot shows the results of the first two principal components being separated mainly due to the presence of FA, VAN A, Iso FA, and CAF. They are the responsible bound form of phenolic compounds to have a large positive loading on component 1. Therefore, genotypes of AAC Synergy, AAC Goldman, AC Metcalfe, CDC Mindon, and Line 1 have a positive relationship due to the presence of high FA, VAN A, Iso FA, and CAF content. CAT, PCA and FA have positive loadings on component 2, whereas genotypes CDC Copeland and Line 4 have a relationship due to the presence of CAT, PCA and FA (Figure 5A–C).

Cluster analysis generated a dendrogram, as illustrated in Figure 5. Different colours indicate different groupings of the barley genotypes based on their bound phenolic compounds’ similarity level at 85%. The analysis is based on the mean values of the eight measured bound forms of the phenolic compound concentrations in 14 barley genotypes into clusters based on a similarity level of 80%. Each cluster was markedly different from the other clusters and consisted of various types of barley genotypes.

In the dendrogram, it shows there is a similarity between AAC Goldman and Line 1 shown in red colour at a level of 96.54% similarity in cluster 2, and the cluster centroids show that it is because of the presence of PCA, FA and CAT. Cluster 3 which is shown in purple colour has another two genotypes; CDC Bow and Line 3 that have a similarity level of 95.46% due to the positive relationship of CAT, PCA and FA. There is a 92.74% similarity level between CDC Bow and Line 4 in cluster 3 based on CAT and FA. Based on the bound phenolic compound analysis, AAC Synergy, AC Metcalfe, and CDC Copeland can be grouped as one, whereas AAC Goldman, Line 1, and CDC Mindon can be grouped as another group of genotypes. The genotypes CDC Bow, Line 3, Line 4, and Line 5 also belong to one group having more than 80% similarity level.

### 2.5. Antioxidant Activity Assay Analysis

The antioxidant activity of barley was comparatively high and was not significantly different between genotypes, although the lowest activity was observed in elite breeding Line 5 (Figure 6). There was also no significant correlation between the antioxidant activity measured by ABTS assay and any of the relevant individual phenolic compounds in the extract (Table 5). However, the correlation matrix showed a strong relationship between some compounds, such as caffeic acid and isoferulic acid, as well as ferulic acid and isoferulic acid.

## 3. Discussion

Barley is a staple food in some countries, and the nutritional constituents of barley include different types of phytochemicals such as phenolic acids, flavonoids, lignans, vitamin E, sterols, and folates [24]. These bioactive phytochemicals are known to have health-promoting benefits [3]. It has been suggested that the total phenolic compound content of barley grains is concentrated in the bran rather than the endosperm [25]. The content of bound insoluble phenolics was higher than those of soluble conjugate and free phenolic extractions, similar to the results observed in our study. The content of barley phenolics may also vary based on the extraction method and the genotype or cultivar. For example, protocatechuic acid, chlorogenic acid, gallic acid, and syringic acid were not detected in any of our tested hulled Canadian genotypes, though they were detected in previous studies [26] when hulless, colored (blue) barley varieties from the Qinghai-Tibet Plateau were assessed. A comparison of phenolic compounds in various barley genotypes showed that CAT is the major phenolic compound in free phenolics of barley, similar to that observed in previous studies [26,27]. Similarly, FA and PCA were the two major polyphenols representing more than 70% of the total bound phenolics in all genotypes studied, while caffeic acid, sinapic acid, isoferulic acid, catechin, 4HBA, vanillic acid, and vanillin were detected in smaller quantities in our study.

The contents of barley lipids could be changed based on the cultivars used [13]. Indeed, the FAMEs analysis of our study showed a variation among the tested 14 Canadian barley genotypes. Linoleic acid, linolenic acid, and oleic acid were among the unsaturated fatty acids involved in reducing cardiovascular risk by decreasing the LDL-cholesterol level [14]. Linoleic acid content (C18:2) in barley grains is the highest and represents about 50%, followed by oleic acid (C18:1) with about 11% and linolenic acid (C18:3) with about 4%, which is similar to that reported previously [13]. Citric acid trimethyl ester and methyl 9(z), 11 (E), 13 (E) octadecatrienoate or conjugated linolenic acid (CLnA), which have anticancer activity, were detected in trace amounts by GC-MS analysis [28]. The existence of citric acid trimethyl ester was only reported in only a few studies in barley [29,30].

Besides growth and environmental conditions, the barley protein content could vary based on the cultivar [12]. A previous study [31] revealed that the protein content in the barley samples fluctuated from 10–17%. However, based on another study [24], the total protein content was from 12 to 16%. Interestingly, the protein content of the germplasm tested in our study was slightly lower, ranging from 7 to 12%. Ascorbic acid was only detected in soluble free barley extracts. The antioxidant capacities of barley were highly similar to that reported in previous research [32], although there were no significant differences between the genotypes. It may be useful to use several assays with different modes of action to evaluate the antioxidant activity of the extract in vitro. The relationships between some phenolic compounds, such as caffeic acid and isoferulic acid or ferulic acid and isoferulic acid, might be due to synergistic effects in their response to environmental conditions that may need further investigation.

Overall, the results of our study showed varying quantities of phenolics, fatty acids, proteins, ascorbic acid content, and antioxidant activity in tested Canadian barley genotypes. The outcome of this study could be used as a potential genotype distinguishing factor(s).

## 4. Materials and Methods

Grains of several two-row spring hulled barley cultivars and elite breeding lines (malting and general purpose) that were grown during the 2020 crop season in the field of the Brandon Research and Development Centre, Brandon, MB, Canada, were used in this study (Table 1).

### 4.1. Preparation of Samples for the Extraction and Analysis

Grains of each of the genotypes included in the study were milled by a Perten Labmill 3100, PerkinElmer Inc. (Woodbridge, ON, Canada), and subsamples of the obtained homogenous powder was subjected to oven drying at 60 °C for 24 h to alleviate any fluctuations in moisture before GC/MS and high-performance liquid chromatography (HPLC) analysis.

### 4.2. Chemical Standards

Chemical standards (gallic acid, protocatechuic acid, ascorbic acid, caffeic acid, chlorogenic acid, p-coumaric acid, ferulic acid, syringic acid, vanillic acid, vanillin, sinapic acid, catechin, isoferulic acid, 4 hydroxybenzoic acid, and FAMEs mix) were purchased from Millipore Sigma (St. Louis, MO, USA). All solvents used were purchased from Fisher Scientific (Waltham, MA, USA) and were HPLC grade.

### 4.3. Extraction and Analysis of FAMEs by GC/MS

Fatty acid analysis was performed using a one-step extraction/methylation method adapted from a previous study [33] with some modifications. Accurately weighed 200 mg of whole meal barley flour was placed in Pyrex glass tubes. Two milliliters of 3N methanolic-HCl were added, and the tubes were incubated in a digital dry bath (Boekel Scientific, PA, USA) at 80 °C for 45 min. Four milliliters of hexane were added to extract the fatty acids methyl esters (FAMEs), vortexed and kept overnight in dark conditions. The hexane layer was transferred using a glass Pasteur pipette to new glass, disposable tubes and evaporated under a vacuum to dryness (60 °C for 40 min) using Eppendorf Vacufuge Plus (Eppendorf, Germany). The extract was suspended in 700 μL hexane and vortexed for 30 s, and transferred to GC vials prior to analysis. An analysis of the FAMEs was performed using Bruker 436-GC equipped with EVOQ-TQ-MS (Bruker Daltonics, Germany). FAMEs were separated on an Rt-2560 capillary column (100 m × 250 mm ID × 0.20 um df) (Restek, Bellefonte, PA, USA). The carrier gas was ultrapure Helium with an initial flow rate of 0.8 mL/min. The temperature for the inlet was 250 °C, and the GC temperature program was set at 100 °C for 4 min ramping up to 250 °C at a rate of 3 °C/min and holding for 8 min. The MS was operated in the electron ionization mode at 70 eV in the full scan mode in the range of 50 to 500 amu using MSWS software. FAMEs were identified based on comparing their retention times with the retention times of a standard mix as well as searching the NIST library. FAMEs were represented in tables as a percentage relative to the total percentage.

### 4.4. Protein Composition Analysis

The grain protein of the samples was measured by a combustion nitrogen analysis (CNA) using a LECO Model FP-628 CNA (St. Joseph, MI, USA) analyzer calibrated by ethylenediamine tetraacetic acid (EDTA). Prior to that, the samples were ground on a UDY Cyclone Sample Mill fitted with a 0.5-mm screen (UDY Corporation, Fort Collins, CO, USA), and a moisture analysis was performed. The results were reported on a dry matter basis (ASBC Barley 7C).

### 4.5. Determination of Ascorbic Acid (Vitamin C)

The HPLC method employed a simultaneous analysis of free phenolics and ascorbic acid, an organic acid, in the same run. Extraction of ascorbic acid was performed by ethanol 90% solution using the same protocol for extracting the free phenolics as described earlier. Quantification of ascorbic acid was based on an external standard calibration method. A serial dilution of concentrations ranging from 24.1 to 386 μg/mL in methanol was prepared and analyzed by HPLC, and a standard curve was established between the concentrations and corresponding peak areas. The resulting equation was used to determine the concentration of ascorbic acid in sample extracts.

### 4.6. Analysis of Free Phenolic Compounds in Barley Using HPLC

Free phenolic molecules from the barley whole meal flour were extracted from adapted protocols of previously published studies using an ethanol 90% solution with modifications [27,34]. One gram of powder was extracted with 15 mL of ethanol 90% and sonicated in a water bath at 70 °C for 1 h with periodical inverting of tubes. The supernatant was filtered, using Whatman#4 filter paper, into new tubes and concentrated under a vacuum at 60 °C for 2 h using Eppendorf Vacufuge Plus (Eppendorf, Germany). The residue was suspended in 500 μL HPLC-grade methanol, then sonicated for 5 min at 50 °C until complete solubility. This methanolic extract was centrifuged at 12,000× *g* rpm for 20 min, and 140 μL aliquot was transferred into vial inserts prior to HPLC analysis.

### 4.7. Analysis of Bound Phenolic Compounds in Barley Using HPLC

Extraction of bound phenolic compounds from barley whole meal flour was also adopted from a previous study [35] with modifications and scaling down to be more environmentally friendly. It was based on successive acid and alkaline hydrolysis. Accurately weighed 0.1 g powder was weighed in 2 mL Eppendorf tubes and subjected to acid hydrolysis (1 mL, 2 N HCl). Tubes were inverted several times and sonicated in a water bath for 30 min at 60 °C. After cooling down, tubes were centrifuged at 12,000× *g* rpm for 15 min (Sorvall Biofuge Primo (Thermo Scientific, Waltham, MA, USA) and the supernatant was collected and discarded. Tubes containing the solid residues were let dry in a fume hood for 30 min then 1 mL of ddH_2_O was added to remove the acid residue. Tubes were centrifuged at 12,000× *g* rpm for 15 min, and the supernatant was discarded while the solid residues were left overnight to dry. Alkaline hydrolysis was performed by adding 600 μL 2N NaOH to tubes, vortexed and exposed to a sonication water bath for 30 min at 60 °C and tubes were inverted periodically. After cooling down, the pH of the tubes was adjusted to 2 with concentrated HCl. Extraction of bound phenolics was performed after adding 800 μL ethyl acetate to each tube, vortexing for 30 s and centrifuging at 12,000× *g* rpm for 15 min, and the supernatant was collected in new Eppendorf tubes. The second extraction was performed with 500 μL ethyl acetate, and the supernatants were pooled and concentrated under vacuum at 60 °C for 25 min using Eppendorf Vacufuge Plus (Eppendorf, Germany) until complete dryness. Two hundred microliters of HPLC-grade methanol were added to each tube, vortexed, and centrifuged at 12,000× *g* rpm for 20 min. An aliquot of 100 μL was placed in a vial insert and analyzed with HPLC. An analysis of compounds was performed using HPLC Dionex 3000 Ultimate (Thermo Scientific) equipped with a C18 reversed-phase column (Acclaim 120, 4.6 × 250 mm, 5 μm). Gradient elution was used to separate the compounds, where mobile phase A consisted of acidified water (0.1% phosphoric acid) and mobile phase B was acetonitrile. This gradient started with 10% B increasing to 90% in 38 min, and then equilibrated to the initial concentration before the next injection with a flow rate of 1 mL/min and injection volume of 20 μL. Compounds were acquired at different wavelengths, but the quantification of cinnamic acid derivatives was performed at 325 nm, while benzoic acid derivatives and flavonoids were quantified at 280 nm. The method of external standard calibration was used for the quantification of each compound after establishing the calibration curves covering at least 5 concentration points. The regression equation was used to quantify each compound in the unknown sample, and the results were expressed as μg/mg dry weight. Figure 7 shows the elution order of a mix of standards detected in barley extract.

### 4.8. Antioxidant Activity Assay in Barley

The barley extracts used in the bound phenolic compound analysis study were diluted one time by methanol. The analysis was performed according to a previous study [36]. After the addition of a reaction mixture (supplied in ABTS Antioxidant Assay Kit from amsbio, Cambridge, MA, USA), samples were incubated on a plate shaker at room temperature and read absorbance using a plate reader (FLUOstar Omega, BMG LABTECH, Ortenberg, Germany) at a wavelength of 405 nm. Data were expressed as Trolox equivalents (TE) per gram of starting sample (i.e., μM TE/g). All assays were run in triplicates.

### 4.9. Statistical Analysis

Quantitative data were subjected to one-way ANOVA using OriginPro 2022 (OriginLab Corporation, Northampton, MA, USA). Tukey’s test was used to compare the means between cultivars or elite germplasm for each compound at *p* ≤ 0.05 level. A principal component analysis (PCA) was used to examine the contents of phenolic compounds in each genotype, followed by a cluster analysis.

## 5. Conclusions

Out of the genotypes tested in this study, CDC Copeland and CDC Bold are the genotypes that contained high values of phenolic compounds, fatty acids, proteins, ascorbic acid, and antioxidant activity. Elite breeding Line 5 has the lowest free phenolics, lowest antioxidant activity, and the second highest protein content. Due to the diversity and the quantitative differences of the parameters tested in these genotypes, we suggest that the contents of phenolic compounds, fatty acids, proteins, ascorbic acid, and antioxidant activity could be used as potential barley cultivar distinguishing factors. They will be beneficial to the breeders within their breeding programs and farmers during the cultivation of these cultivars. Moreover, they could serve for the assurance of purity during commercialization and even maybe for cultivar registration and protection. While our results are very interesting, further investigation on a larger number of barley cultivars and elite breeding lines is needed as well as across various growing conditions and years of testing.

## Figures and Tables

**Figure 1 molecules-27-07852-f001:**
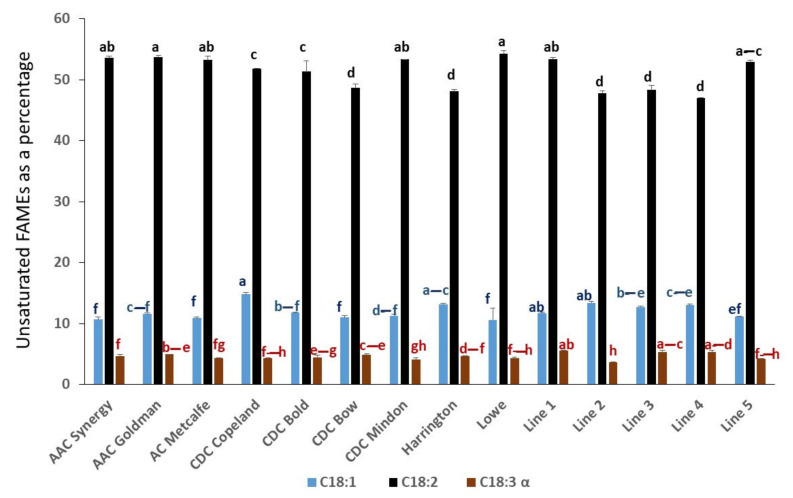
GC/MS analysis of barley cultivars and elite germplasm. Unsaturated FAMEs composition in grain samples is shown. Values sharing the same letter/s in the bars of the same color are not significantly different at the level ≤ 0.05 using Tukey test.

**Figure 2 molecules-27-07852-f002:**
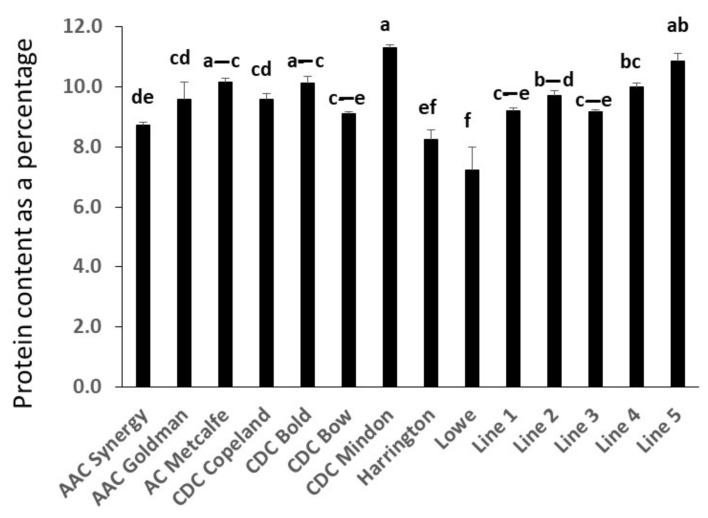
Protein content in tested barley grain samples assessed by LECO protein analyzer. Error bars represent the standard deviation of three replicates each. Values sharing the same letter/s are not significantly different at the level ≤ 0.05 using Tukey test.

**Figure 3 molecules-27-07852-f003:**
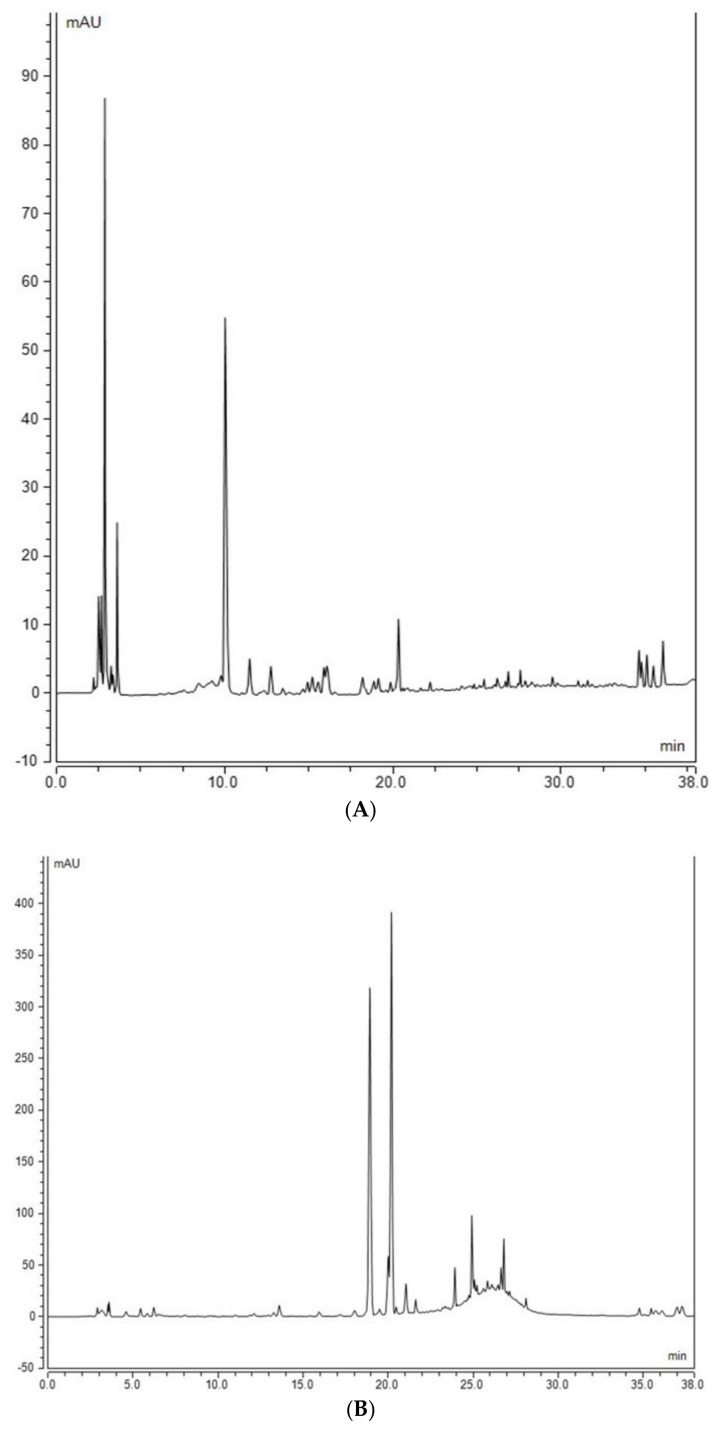
HPLC chromatograms of free and bound compounds in the extracts of elite breeding Line 5 at 280 nm. (**A**) HPLC chromatogram showing the simultaneous analysis of ascorbic acid and free phenolics in the extract of elite breeding Line 5. Compounds identified: Rt 2.867, ascorbic acid; Rt 10.023, unknown; Rt 18.873, PCA; Rt 20.327 FA. (**B**) HPLC chromatogram of the bound phenolics identified in the extract of Line 5. Compounds identified: Rt13.273, VAN A; Rt 13.600, CAF; Rt 18.923, PCA; Rt 20.003, SIN; Rt 20.190, FA; Rt 21.047, IFA.

**Figure 4 molecules-27-07852-f004:**
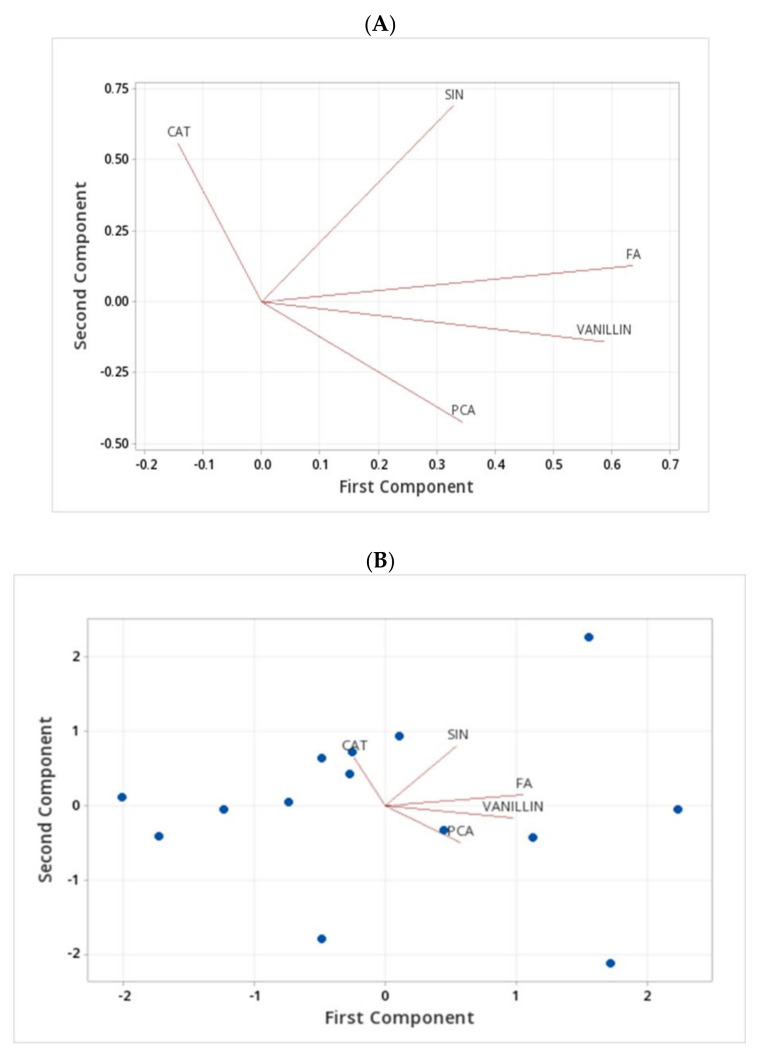
Principal component analysis and cluster analysis of the free form of phenolic compounds in barley genotypes from Table 1; (**A**,**B**) the loading plot: blue dots indicate each barley genotype tested; (**C**) the scree plot of principal component analysis for FA, PCA, CAF, SIN, Iso FA, CAT, 4HBA, and VAN A; (**D**) dendrogram from cluster analysis for FA, PCA, CAF, SIN, Iso FA, CAT, 4HBA, and VAN A of 14 barley genotypes. Observations left to right in order: 1 = AAC Synergy, 2 = AAC Goldman, 3 = AC Metcalfe, 6 = CDC Bow, 11 = Line 2, 4 = CDC Copeland, 7 = CDC Mindon, 5 = CDC Bold, 8 = Harrington, 10 = Line 1, 12 = Line 3, 13 = Line 4, 9 = Lowe, and 14 = Line 5.

**Figure 5 molecules-27-07852-f005:**
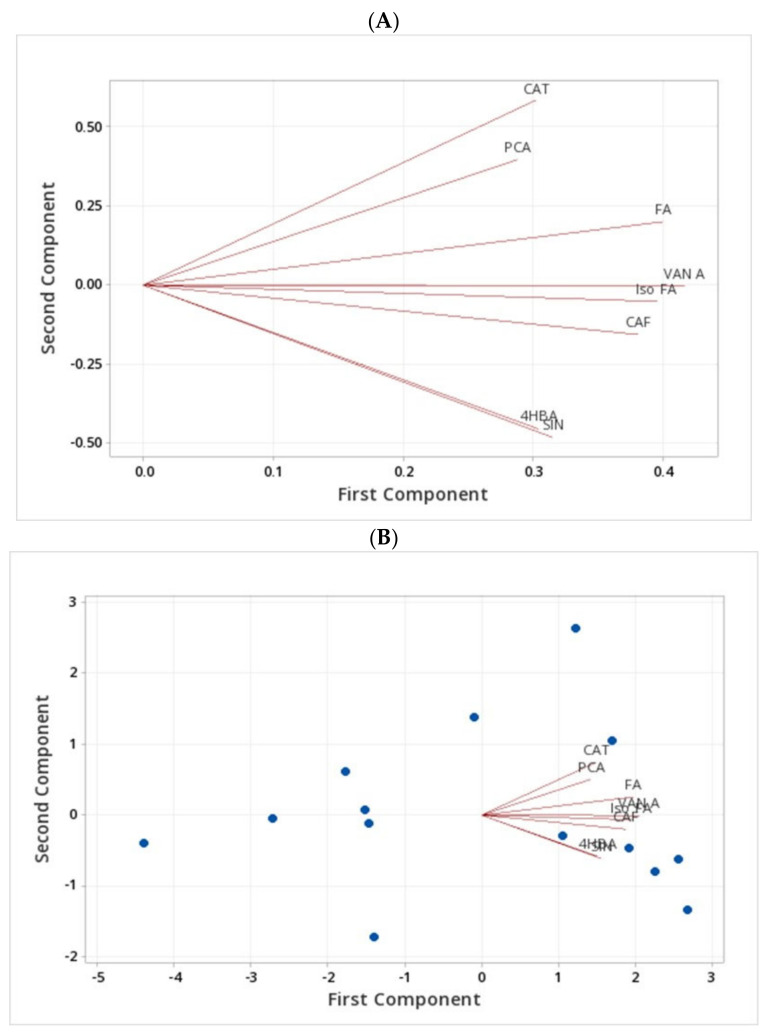
Principal component analysis and cluster analysis of the bound form of phenolic compounds in barley genotypes from Table 1; (**A**,**B**) the loading plot: blue dots indicate each barley genotype tested; (**C**) the scree plot of principal component analysis for FA, PCA, CAF, SIN, Iso FA, CAT, 4HBA, and VAN A; (**D**) dendrogram from cluster analysis for FA, PCA, CAF, SIN, Iso FA, CAT, 4HBA, and VAN A of 14 barley genotypes. Observations left to right in order: 1 = AAC Synergy, 3 = AC Metcalfe, 4 = CDC Copeland, 2 = AAC Goldman, 10 = Line 1, 7 = CDC Mindon, 6 = CDC Bow, 12 = Line 3, 13 = Line 4, 14 = Line 5, 9 = Lowe, 8 = Harrington, 5 = CDC Bold, and 11 = Line 2.

**Figure 6 molecules-27-07852-f006:**
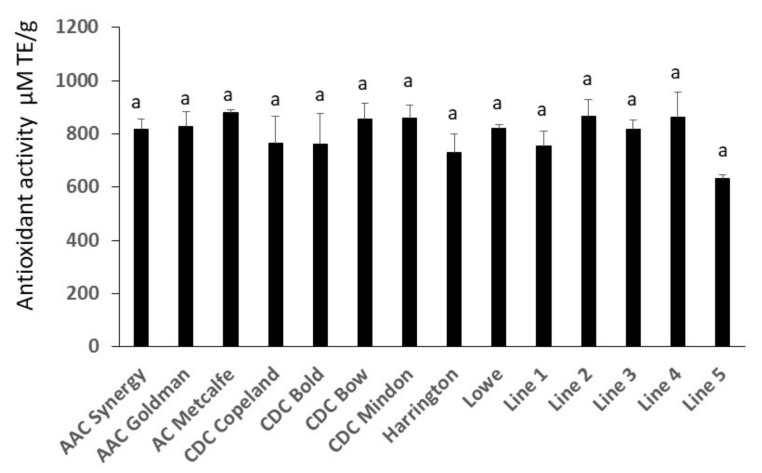
Antioxidant activity in tested barley grain samples. Data are expressed as Trolox equivalents (TE) per gram of the starting sample (i.e., μM TE/g). Error bars represent the standard deviation of three replicates each. Values sharing the same letter are not significantly different at the level ≤ 0.05 using Tukey test.

**Figure 7 molecules-27-07852-f007:**
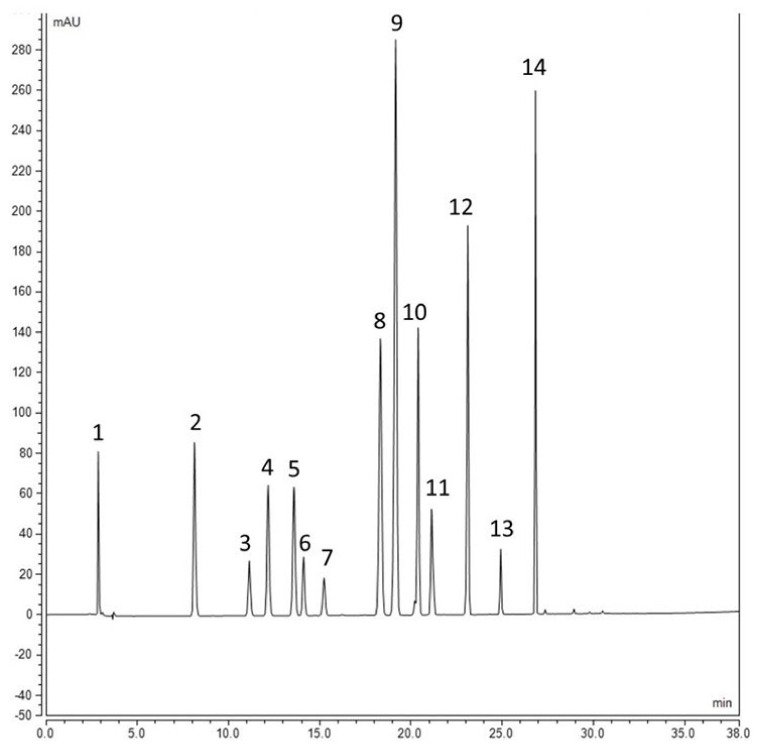
HPLC chromatogram of the standards tested. 1—Rt 2.857, Ascorbic acid; 2—Rt 8.123, Protocatechuic acid; 3—Rt 11.130, Catechin; 4—Rt 12.163, 4-Hydroxybenzoic acid (4HBA); 5—Rt 13.580, Vanillic acid; 6—Rt 14.107, Epicatechin; 7—Rt 15.227, 3HBA; 8—Rt 18.317, Vanillin; 9—Rt 19.163, Para coumaric acid; 10—Rt 20.390, Ferulic acid; 11—Rt 21.123 Iso ferulic acid; 12—Rt 23.110 O-coumaric acid; 13—Rt 24.920 Salicylic acid; 14—Rt 26.833, Genistein.

**Table 1 molecules-27-07852-t001:** List of the two-row spring hulled cultivars and elite breeding lines used in the study.

#	Name	Class	Registration	Reference
1	AAC Synergy	Malting	2012	[15]
2	AAC Goldman	Malting	2018	[16]
3	AC Metcalfe	Malting	1997	[17]
4	CDC Copeland	Malting	1999	[18]
5	CDC Bold	General purpose	1999	[19]
6	CDC Bow	Malting	2016	[20]
7	CDC Mindon	General purpose	2007	[21]
8	Harrington	Malting	1981	[22]
9	Lowe	Malting	2016	[23]
10	Line 1	intended for Malting	na	na
11	Line 2	intended for Malting	na	na
12	Line 3	intended for Malting	na	na
13	Line 4	intended for Malting	na	na
14	Line 5	intended for Malting	na	na

**Table 2 molecules-27-07852-t002:** Ascorbic acid (ASA) content (μg/g dwt) in tested barley grain samples.

Genotypes	ASA CONTENT (μg/g dwt)
AAC Synergy	361 ± 12 ^b^
AAC Goldman	259 ± 22 ^b^
AC Metcalfe	366 ± 40 ^b^
CDC Copeland	460 ± 21 ^a^
CDC Bold	522 ± 91 ^a^
CDC Bow	350 ± 96 ^b^
CDC Mindon	453 ± 48 ^a^
Harrington	425 ± 28 ^a^
Lowe	425 ± 38 ^a^
Line 1	354 ± 62 ^b^
Line 2	332 ± 3.5 ^b^
Line 3	340 ± 15 ^b^
Line 4	407 ± 30 ^a^
Line 5	307 ± 11 ^b^

Values sharing the same letter/s in the column are not significantly different at the level ≤ 0.05 using Tukey test.

**Table 3 molecules-27-07852-t003:** Free phenolic compound content (μg/g dwt) in barley grain samples.

Genotypes	Phenolic Compound Content (μg/g dwt)				
	PCA	SIN	FA	CAT	VANILLIN
AAC Synergy	ND	0.23 ± 0.01 ^cd^	2.19 ± 0.1 ^c^	15.8 ± 1.1 ^bc^	0.88 ± 0.03 ^b^
AAC Goldman	ND	0.19 ± 0.1 ^cd^	2.00 ± 0.1 ^cd^	16.1 ± 1.4 ^bc^	0.87 ± 0.04 ^b^
AC Metcalfe	ND	0.17 ± 0.08 ^cd^	1.79 ± 0.1 ^d^	13.3 ± 1.0 ^c^	0.86 ± 0.06 ^b^
CDC Copeland	0.056 ± 0.02 ^b^	0.34 ± 0.2 ^cd^	2.48 ± 0.2 ^ab^	18.3 ± 0.6 ^b^	0.82 ± 0.11 ^b^
CDC Bold	0.109 ± 0.04 ^b^	0.36 ± 0.1 ^ce^	2.50 ± 0.2 ^ab^	26.9 ± 0.9 ^a^	0.86 ± 0.02 ^b^
CDC Bow	ND	0.11 ± 0.04 ^d^	1.79 ± 0.2 ^d^	20.1 ± 5.1 ^b^	0.85 ± 0.12 ^b^
CDC Mindon	ND	0.42 ± 0.2 ^c^	2.61 ± 0.1 ^ab^	15.5 ± 1.0 ^bc^	0.80 ± 0.05 ^b^
Harrington	ND	0.35 ± 0.04 ^cd^	2.78 ± 0.1 ^a^	13.8 ± 1.0 ^c^	1.11 ± 0.09 ^a^
Lowe	0.283 ± 0.03 ^a^	0.25 ± 0.1 ^cd^	2.44 ± 0.1 ^ab^	8.1 ± 0.5 ^d^	0.95 ± 0.10 ^ab^
Line 1	0.102 ± 0.07 ^b^	0.31 ± 0.1 ^cd^	2.37 ± 0.2 ^bc^	16.3 ± 1.3 ^bc^	1.01 ± 0.08 ^ab^
Line 2	ND	0.71 ± 0.04 ^b^	1.97 ± 0.1 ^cd^	13.5 ± 0.1 ^c^	0.91 ± 0.06 ^ab^
Line 3	ND	0.23 ± 0.006 ^cd^	2.40 ± 0.1 ^ab^	12.3 ± 0.3 ^cd^	0.97 ± 0.02 ^ab^
Line 4	ND	1.48 ± 0.1 ^a^	2.52 ± 0.1 ^ab^	13.2 ± 1.0 ^c^	0.94 ± 0.01 ^ab^
Line 5	ND	ND	2.28 ± 0.1 ^b^	ND	0.87 ± 0.15 ^b^

PCA, para coumaric acid; SIN, sinapic acid; FA, ferulic acid; CAT, catechin; vanillic acid; VANILLIN, vanillin. Values sharing the same letter/s in the column are not significantly different at the level ≤ 0.05 using Tukey test.

**Table 4 molecules-27-07852-t004:** Bound phenolic compound content (μg/g dwt) in barley grain samples.

Genotypes	Phenolic Compound (μg/g dwt)								
	PCA	CAF	SIN	FA	Iso FA	CAT	4HBA	VAN A	VANILLIN
AAC Synergy	214 ± 22 ^ab^	12.8 ± 1.0 ^ab^	19.8 ± 0.9 ^a^	382 ± 14.3 ^a^	22.5 ± 1.0 ^ab^	20.3 ± 2.0 ^ab^	2.90 ± 0.32 ^a^	9.21 ± 1.21 ^a^	7.99 ± 1.13 ^ab^
AAC Goldman	245 ± 54 ^a^	14.5 ± 0.8 ^a^	20.0 ± 2.9 ^a^	377 ± 33.2 ^a^	21.8 ± 1.4 ^ab^	18.0 ± 2.9 ^ab^	3.18 ± 0.12 ^a^	9.33 ± 1.41 ^a^	7.39 ± 0.78 ^ab^
AC Metcalfe	202 ± 23 ^b^	11.1 ± 0.8 ^ab^	14.4 ± 0.1 ^b^	338 ± 15.4 ^a^	20.9 ± 1.3 ^ab^	19.2 ± 3.3 ^ab^	2.81 ± 0.09 ^a^	9.88 ± 1.71 ^a^	6.32 ± 1.02 ^ab^
CDC Copeland	236 ± 34 ^ab^	10.6 ± 2.1 ^ab^	15.2 ± 2.5 ^a^	342 ± 4.2 ^a^	16.9 ± 2.9 ^ab^	21.5 ± 7.1 ^a^	1.64 ± 0.24 ^ab^	8.19 ± 1.49 ^ab^	7.25 ± 0.76 ^ab^
CDC Bold	140 ± 17 ^b^	8.6 ± 2.6 ^b^	8.9 ± 1.9 ^b^	191 ± 46.5 ^b^	12.2 ± 4.1 ^b^	12.8 ± 0.8 ^b^	1.81 ± 0.27 ^b^	4.85 ± 0.25 ^b^	5.56 ± 0.71 ^b^
CDC Bow	228 ± 53 ^ab^	9.9 ± 0.7 ^b^	15.7 ± 1.7 ^a^	293 ± 41.5 ^b^	14.2 ± 1.8 ^ab^	15.9 ± 2.1 ^ab^	1.88 ± 0.07 ^b^	6.74 ± 1.23 ^b^	5.58 ± 0.74 ^b^
CDC Mindon	265 ± 35 ^a^	15.8 ± 3.3 ^a^	20.3 ± 3.7 ^a^	375 ± 51.5 ^a^	22.5 ± 5.5 ^ab^	17.5 ± 3.1 ^ab^	2.33 ± 0.11 ^ab^	9.37 ± 1.65 ^a^	8.80 ± 0.91 ^a^
Harrington	209 ± 27 ^ab^	13.5 ± 2.1 ^ab^	13.3 ± 2.9 ^b^	469 ± 101 ^a^	24.2 ± 7.0 ^a^	20.2 ± 1.8 ^ab^	1.88 ± 0.13 ^b^	8.41 ± 0.08 ^ab^	9.13 ± 1.48 ^a^
Lowe	331 ± 96 ^a^	10.4 ± 1.1 ^ab^	11.3 ± 0.8 ^b^	368 ± 25.5 ^a^	17.4 ± 1.5 ^ab^	24.6 ± 2.9 ^a^	2.21 ± 0.25 ^b^	9.41 ± 1.45 ^a^	8.33 ± 1.12 ^ab^
Line 1	239 ± 30 ^a^	11.8 ± 0.4 ^ab^	21.0 ± 1.0 ^a^	370 ± 17.5 ^a^	21.7 ± 0.9 ^ab^	19.6 ± 2.8 ^ab^	2.52 ± 0.19 ^ab^	9.37 ± 0.99 ^a^	7.70 ± 0.70 ^ab^
Line 2	116 ± 18 ^b^	10.1 ± 2.0 ^ab^	10.2 ± 2.1 ^b^	273 ± 64.0 ^b^	17.7 ± 6.1 ^ab^	15.2 ± 4.6 ^ab^	1.54 ± 0.19 ^b^	6.19 ± 1.07 ^b^	5.86 ± 1.36 ^b^
Line 3	216 ± 23 ^ab^	9.7 ± 1.4 ^b^	15.6 ± 2.2 ^a^	297 ± 8.6 ^b^	13.7 ± 2.2 ^ab^	16.8 ± 4.2 ^ab^	2.21 ± 0.26 ^b^	6.45 ± 1.14 ^b^	6.41 ± 0.22 ^ab^
Line 4	232 ± 31 ^ab^	10.0 ± 3.7 ^b^	11.2 ± 2.5 ^b^	310 ± 60.8 ^b^	14.6 ± 5.3 ^ab^	15.2 ± 2.1 ^ab^	1.83 ± 0.05 ^b^	6.76 ± 1.72 ^b^	7.14 ± 0.87 ^ab^
Line 5	191 ± 16 ^b^	9.3 ± 1.7 ^b^	17.7 ± 1.2 ^a^	294 ± 53.2 ^b^	14.1 ± 1.7 ^ab^	10.5 ± 2.5 ^b^	2.39 ± 0.27 ^ab^	8.26 ± 0.28 ^ab^	6.69 ± 0.18 ^ab^

PCA, para-coumaric acid; CAF, caffeic acid; SIN, sinapic acid; FA, ferulic acid; Iso FA, isoferulic acid; CAT, catechin; 4HBA, 4-hydoxybenzoic acid; VAN A, vanillic acid; VANILLIN, vanillin. Values sharing the same letter/s in the column are not significantly different at the level ≤ 0.05 using Tukey test.

**Table 5 molecules-27-07852-t005:** Correlation matrix depicting the relationship between the antioxidant activity (ABTS) assay and individual bound phenolic compounds in barley cultivars.

	Antioxidant	PCA	CAF	SIN A	FA	Iso FA	CAT	4HBA	VAN A	VANILLIN
Antioxidant		0.44	0.22	0.095	0.04	0.209	0.034	0.204	0.166	−0.23
PCA			0.334	0.33	0.551 *	0.244	0.691 **	0.078	0.607 *	0.534 *
CAF				0.587 *	0.736 **	0.859 ***	0.368	0.387	0.56 *	0.691 **
SIN A					0.436	0.5	0.117	0.554 *	0.572 *	0.382
FA						0.837 ***	0.666 **	0.325	0.723 **	0.797 ***
Iso FA							0.519	0.591 *	0.723 **	0.686 **
CAT								0.262	0.584 *	0.486
4HBA									0.626 *	0.064
VAN										0.541 *
VANILLIN										

* Significant at *p* ≤ 0.05; ** significant at *p* ≤ 0.01; *** significant at *p* ≤ 0.001.

## Data Availability

Not applicable.

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
