# Peer review of "Analyses of Fatty Acids, Proteins, Ascorbic Acid, Bioactive Phenolic Compounds and Antioxidant Activity of Canadian Barley Cultivars and Elite Germplasm"

_molecules, 2022, doi:10.3390/molecules27227852_

Round 1

Reviewer 1 Report

 1. Figure 1A, 2 and 4 need to make significant difference analysis and mark different letters.    

2. GC-MS figs like Fig1B can be used as supplementary figs and do not need to be shown in the paper.    

3. Cluster analysis is recommended for different varieties.    

4. Principal component analysis is recommended for data indicators.    

5. It is suggested to conduct correlation analysis between antioxidant activity and compound content.

Author Response

  1. Figure 1A, 2 and 4 need to make significant difference analysis and mark different letters.

Thank you very much for the suggestion.  All the figures mentioned are marked with  letters to represent the significant interactions of the analysis.

  1. GC-MS figs like Fig1B can be used as supplementary figs and do not need to be shown in the paper.

Thank you for the suggestion. The figures are now included in the supplementary material (supplementary figure 1)

  1. Cluster analysis is recommended for different varieties.

Thank you for the suggestion.  Cluster analysis was conducted for the individual phenolic compounds of free and bound forms, and they were now presented in Figures 4D and 5D.

  1. Principal component analysis is recommended for data indicators.

Thank you for the suggestion. Principal component analysis was conducted for the individual phenolic compounds of free and bound fractions, and is now presented in Figures 4 A to C and Figure 5 A to C.

  1. It is suggested to conduct correlation analysis between antioxidant activity and compound content.

Thank you for the suggestion. Correlation analysis was done for the 14 barley genotypes to grasp the differences in free and bound phenolic contents. The results of the analysis are presented in Table 5.

Overall, we were able to address all the suggestions made by reviewer 1.  We thank the reviewer for these excellent suggestions.

Reviewer 2 Report

In this study, authors performed a comparative analysis of fatty acids, proteins, ascorbic acid, phenolic 15compounds and antioxidant activity of several Canadian barley cultivars and elite breeding lines.

1.       Rationale of the study needs to be incorporated.

2.       correlation analysis should be added.

3.       Figure representation is very poor.

4.       unnecessary reference to literature.

5.       Please standardize the way of writing the percentage i.e. protein contents (%).

6.       Data presentation needs to be improved.

7.       English correction is compulsory.

8.       I think, there is a mistake in title. It should be corrected.

Author Response

In this study, authors performed a comparative analysis of fatty acids, proteins, ascorbic acid, phenolic 15compounds and antioxidant activity of several Canadian barley cultivars and elite breeding lines.

  1. Rationale of the study needs to be incorporated.

We have elaborated the rationale in the abstract (Line 17) and the introduction (Line 71). Thank you very much for the suggestion.

  1. correlation analysis should be added.

Thank you for the suggestion. A correlation analysis was conducted between antioxidant activity and individual phenolic compounds. The results are presented in Table 5 and included in the text.

  1. Figure representation is very poor.

Thank you for the constructive criticism. We have added significant interactions based on the statistical analysis in figures (Figure 1 & 2), and additional figures on PCA analysis (Figure 4 A to C & 5 A to C) and cluster analysis (Figure 4 D & 5D).

  1. unnecessary reference to literature.

Thank you for your feedback. we have checked and removed the references which were unnecessary.

  1. Please standardize the way of writing the percentage i.e. protein contents (%).

We have done as requested.

  1. Data presentation needs to be improved.

We have revised the figures (including statistical analysis letters) and added new figures (Figure 4 and 5) and tables. Thank you for the suggestion.

  1. English correction is compulsory.

Dr. Netticadan (co-author) is a native English speaker, he has served as editor of special issues. Dr. Netticadan has checked and edited the final manuscript.

  1. I think, there is a mistake in title. It should be corrected.

We have edited the title to “Analyses of fatty acids, proteins, ascorbic acid, bioactive phenolic compounds and antioxidant activity of Canadian barley cultivars and elite germplasm” to better fit the contents of the manuscript.

We thank the reviewer for his/her comments. The suggestions were addressed in the revised version of the manuscript.

Round 2

Reviewer 2 Report

NA